# Antibiotic Use in Livestock and Residues in Food—A Public Health Threat: A Review

**DOI:** 10.3390/foods11101430

**Published:** 2022-05-16

**Authors:** Oana Mărgărita Ghimpețeanu, Elena Narcisa Pogurschi, Dana Cătălina Popa, Nela Dragomir, Tomița Drăgotoiu, Oana Diana Mihai, Carmen Daniela Petcu

**Affiliations:** 1Faculty of Veterinary Medicine, University of Agronomic Sciences and Veterinary Medicine of Bucharest, 050097 Bucharest, Romania; ghimpe_marga@yahoo.com (O.M.G.); oprea_diana2008@yahoo.com (O.D.M.); 2Faculty of Animal Productions Engineering and Management, University of Agronomic Sciences and Veterinary Medicine of Bucharest, 011464 Bucharest, Romania; elena.pogurschi@usamv.ro (E.N.P.); dana-catalina.popa@igpa.usamv.ro (D.C.P.); nela.dragomir@usamv.ro (N.D.); tomita.dragotoiu@usamv.ro (T.D.)

**Keywords:** antibiotic residues, food safety, antimicrobial resistance, public health

## Abstract

The usage of antibiotics has been, and remains, a topic of utmost importance; on the one hand, for animal breeders, and on the other hand, for food safety. Although many countries have established strict rules for using antibiotics in animal husbandry for the food industry, their misuse and irregularities in compliance with withdrawal periods are still identified. In addition to animal-origin foods that may cause antibiotic residue problems, more and more non-animal-origin foods with this type of non-compliance are identified. In this context, we aim to summarize the available information regarding the presence of antibiotic residues in food products, obtained in various parts of the world, as well as the impact of consumption of food with antibiotic residues on consumer health. We also aim to present the methods of analysis that are currently used to determine antibiotic residues in food, as well as methods that are characterized by the speed of obtaining results or by the possibility of identifying very small amounts of residues.

## 1. Introduction

Antibiotics are produced by living organisms or in the laboratory and have the capacity to kill or inhibit the growth of microorganisms [1]. Even if antibiotics represent a positive aspect for both animals and humans, because of their great impact on their health status, their abusive use can lead to harmful consequences, and especially to the appearance of resistant germs [2,3,4].

The practice of using antibiotics is still widespread for prophylactic and curative purposes, and the problem of their remnants in animal products, groundwater, soil, and feed has caused (and is still causing) worldwide concern, generating vast costs to combat antibiotic resistance [5]. In order to reduce this effect, a number of developed countries have enforced laws in order to decrease the excessive use of antibiotics as a means to prevent diseases or as an acceleration tool in animals [6]. A number of monitoring programs regarding the use of antibiotics in animal breeding have been implemented over the last 30 years, starting with the Danish Integrated Antimicrobial Resistance Monitoring and Research Program in 1995 [7]. Nowadays, a number of research projects have been developed in this area; as an example, the Disseminating Innovative Solutions for Antibiotic Resistance Management project (DISARM), in which nine European countries are involved [8]. As a counterbalance, in less developed countries in Asia or Africa, the use of antibiotics is still very high due to the great demand for animal products, and this leads to the uncontrolled use of these substances in the breeding systems [9].

Most antibiotic residues can be found in a large range of food products, both animal and vegetal [10]. Although meat consumption is decreasing in developed countries, especially in Europe, in less developed countries it is very high, so meat represents the most important source of antibiotics. Primarily, the main source of antibiotics in food is abusive use (overdosing, non-compliance with the withdrawal period) and the use of antibiotic-contaminated water, or inappropriate disposal of animal dung.

The sources of antibiotics in humans are two-fold: first, prescriptions for people (most commonly penicillins, macrolides, and fluoroquinolones), and, second, the substances used in animal breeding (tetracycilies and sulfonamides). Antibiotics can cause serious effects on human health that have led to the introduction of maximum residue limits (MRL) in food safety legislation [11,12]. Long-term exposure is linked to impairment of the immune system, digestive problems caused by the destruction of the intestinal flora, renal problems, and even carcinogenic effects [13].

Yang et al. (2021) and Huang et al. (2020) mentioned that antibiotics have become widespread in the environment due to their extensive and long-term use, influencing both human health and the system, due to the emergence of antibiotic resistance [14,15]. Saeed et al. (2020) addressed the contamination of food with antibiotic residues, mentioning that it is a global problem, due to the improper use of antibiotics. The authors of the current paper also mentioned methods of analysis to identify antibiotics in food [16].

This review focuses on three aspects: (1) antibiotic use in livestock; (2) antibiotic residues in food and methods of detection; and (3) the impact of consumption of food with antibiotic residues on consumer health.

## 2. Antibiotic Use in Livestock

Livestock farming plays a very important role in the agriculture of the European Community. Achieving the best results largely depends on the use of safe and good quality feed. Free movement of safe and good quality food and feed is a key aspect of the internal market, and contributes significantly to the health and well-being of consumers [17].

In terms of animals of economic interest and poultry farms, various active substances are used, including antibiotics, in order to maintain the health of the animals and to have a better yield for breeding. Antibiotics can be administered via feed or by intramuscular injection [18,19]. Medicated feed is an oral route of veterinary medicinal product administration. Medicated feed is a homogeneous mixture of feed and veterinary medicinal products [20].

Regulation (EU) 6/2019 establish rules for the authorization of use of veterinary medicinal products in feeding stuffs, including the manufacture, distribution, advertising, and surveillance of such products [12]. Feed business operators, which handle manufacturing, storing, transporting, or placing medicated feed and intermediate products on the market, must be authorized by the competent authority, in accordance with the authorization system, to ensure both the safety of the feed and the traceability of the products. The labeling of medicated feed stuffs must comply with the general principles set out in Regulation (EC) 767/2009, and are subject to specific labeling requirements, in order to provide users with the information necessary for the correct administration of medicated feed [21]. Such administration must be adequately described in the product information to ensure correct administration and proper dosing of certain veterinary medicinal products, to be administered orally to animals, in feed, or in drinking water, especially in the case of treating groups of animals. The relevant legislation provides the establishment of additional instructions for cleaning of the equipment used in the administration of respective medicinal products, in order to avoid cross-contamination and to reduce resistance to antimicrobials [12]. A large range of synthetic feed additives are included in this category of antibiotics and are used in animal feeding to increase production efficiency and to control different diseases [8].

Recent studies have shown that a significant percentage of all manufacturers add antibiotics to healthy animals’ feed to prevent, rather than cure, diseases [22,23,24,25]. According to Erofeeva et al., livestock accounts for approximately 50% of all antibiotics produced in the world. During the life of an animal, the use of antibiotics can significantly increase a gain in weight by increasing the use of nutrients in the diet, which, in the end, makes it possible to reduce feed costs and shorten the feeding period [25]. Feed manufacturers and authorities establish procedures and instructions for the effective and safe use of authorized and prescribed veterinary medicinal products, for oral administration, other than medicated feeding stuff, such as mixing drinking water with veterinary medicinal products or manually mixing a veterinary medicinal product in feed, which are administered by farmers to food-producing animals. These instructions take into account the scientific recommendations of the European Medicines Agency, established by Regulation (EC) No. 726/2004, on measures to minimize over dosage or under dosage, unintended administration to animals other than those targeted, the risk of cross-contamination, and the release of these products into the environment [12,26]. Homogeneous dispersion of veterinary medicinal product in feed is also essential for the manufacture of safe and effective medicated feed [20].

According to different researchers, reliable data regarding the quantity and patterns of use, dose, and frequency, are not very accurate [27,28,29]. Disease prevention is more effective than treating it. Drug treatment, with antimicrobials in particular, should in no way replace good animal husbandry, biosecurity, and management practices [20]. Excessive weight loss at the beginning of the lactation period leads to the appearance of stress, increasing the susceptibility of animals to diseases, which is why medicated feed has been used for animals for curative and preventive purposes [30]. The US Department of Agriculture noted that approximately 88% of growing swine receive antibiotics in their feed for disease prevention and growth promotion purposes, commonly tetracyclines or tylosin [22]. Some of the most frequently used antibiotics in ruminants are ionophores, a distinctive class of antibiotics that can influence intestinal flora to achieve increased energy and amino acid availability and improved nutrient utilization [22]. Most beef calves in feed lots and some dairy heifers receive this drug routinely in their feed. Ionophores have never been used in humans or therapeutically in animals, because of their specific mode of action. The impact that medicated food has on human health, and implicitly how antibiotic resistance is installed after ingesting food with antibiotics residues, is a topic that we will detail in a later section, but it is worth mentioning that while some bacteria are intrinsically resistant to these drugs, there is currently no evidence to suggest that ionophore resistance is transferable [22,28].

The presence of antibiotics in feed is not desirable due to the antibiotic resistance that the bodies can later acquire. Therefore, the elimination of antibiotics from feed, but improvements in the productivity of pigs and poultry, has been achieved through the use of feed additives, such as omega 3, immunoglobulin, organic and inorganic acids, zinc oxide, yeast derived β glucans, essential oils, prebiotics, probiotics, threonine, cysteine, and herbs and spices [31,32]. On the other hand, the conclusion of another study was that the elimination of prophylactic in-feed antibiotics leads to minor reductions in productive performance and animal health [29].

Advertising for medicated feeding stuffs addressed persons who cannot properly assess the risks associated with their use, and may lead to improper or excessive use of the medicinal product, which may harm public or animal health or the environment [20].

Animal health management is mainly based on disease prevention, compliance with hygiene conditions, and the correct application of disinfection actions. Preventive use of synthetic chemical allopathic medicines is forbitten in order to obtain organic products. In case antibiotics are compulsory, treatment should be carefully managed through minimal use with a high efficiency. In such cases, in order to ensure the integrity of organic production for consumers, an official withdrawal period after the use of such medicinal products requires a longer period of time. In organic production, the use of antibiotics as growth or production stimulants, coccidiostats, hormones, etc., is not permitted [33].

## 3. Antibiotic Residues in Food

Dozens of scientific papers have been published over the years, focusing on antibiotic residues in food and feed. Antibiotics can be naturally produced by living organisms or they can be synthetically produced in a laboratory. The main role of antibiotics is to inhibit or kill microorganism growth [1,34]. Antibiotics can be used as feed additives in livestock in order to maintain animal health; lately, however, there is an increasing attempt to stop this phenomenon as antibiotic residues can be consumed by humans with food of animal origin (meat, milk, eggs, fish, honey, etc.). It should be noted that, not only may food from animal origin have antibiotic residues, but also plant foods, which can be contaminated by soil and water [35,36]. From Table 1, it can be seen that antibiotic residues are found in all foods intended for human consumption, whether they are of animal or non-animal origin. It can be seen that most studies were performed to determine the presence of antibiotic residues in milk, followed by meat, and then by honey. Another thing to note is that research on antibiotic residues started in the 1980s, but the largest number of works in this field appeared in the last 5 years.

In order to ensure food safety for consumers, more and more studies have attempted to find effective and rapid methods for the detection antibiotic residues in feed and food [36]. Consumers are also increasingly interested in consuming quality food, and are increasingly turning to organic products, which provide them with more safety relative to conventional products.

Several studies have shown the presence of antibiotic residues in various types of food (Table 2).

### 3.1. Products of Non-Animal Origin

Although food products of animal origin are considered to be the major source of antibiotic residues, studies in the literature have shown that even non-animal origin products represent an important problem regarding this type of contamination. The main sources of these compounds in agriculture are irrigation water with antibiotics traces due to inappropriate recycling processes or the use of manure as soil amendments, which leads to spreading antibiotics through the food chain [52]. As a result of these agricultural practices, antibiotics in soil can be taken up by plants, entering the food chain. Due to the fact that plants are considered to be a minor source of antibiotic residues, studies have focused on few compounds [29,30,31]. The most common vegetables that accumulate antibiotics are considered to be cereals, such as wheat, rice, and oat, and coarse grains, such as maize and barley. In this field, studies have focused on antibiotic detection in different matrices or were conducted as experiments in a hydroponic environment [53,54,55,56,57].

In recent years, research has been conducted in order to study the relation between intake of antibiotics in edible crops due to poorly management of wastewater used in irrigations and from manure [53,55,58,59,60,61]. Pan and Chu (2017) studied the influence of some antibiotics (tetracycline, sulfamethazine, norfloxacin, erythromycin, and chloramphenicol) on crops in relation to two types of contamination, irrigation with wastewater and soil amendment with animal manure. Findings showed that the distribution of tetracycline, norfloxacin, and chloramphenicol in crop tissues were as follows: fruit > leaf/shoot > root; an opposite order was found for sulfamethazine (SMZ) and erythromycin (ERY), i.e., root > leaf/shoot > fruit [58]. Research also revealed that the uptake of antibiotics in crops was higher in the case of wastewater use and it was lower in the case of manure fertilization, argued by the fact that crops are more likely to absorb residues during the continuous process of irrigation. However, the levels of antibiotics ingested through the consumption of edible crops under the different treatments were much lower than acceptable daily intake (ADI) levels.

In Northern China, studies have shown that oxytetracycline, tetracycline, chlortetracycline, sulfamethoxazole, sulfadoxine, sulfachloropyridazine, chloramphenicol, ofloxacin, pefloxacin, and lincomycin were found in vegetables. In the same geographical area, relatively high concentrations of norfloxacin, ciprofloxacin, and enrofloxacin were found in vegetables, such as tomato, cucumber, pepper, spinach, eggplant, and crown daisy [62]. Other studies have shown that parts of vegetables, such as the roots of carrot and leaves of lettuce, as well as cabbage and spinach, the stem of celery, and fruits of cucumber, bell pepper, and tomato had 64% pharmaceutical residues, including antibiotics, due to the wastewater used for irrigation [63]. Bassil et al. (2013) evaluated the uptake of gentamicin and streptomycin in carrot (*Daucus carota*), lettuce (*Lactuca sativa*), and radish (*Rhaphanus sativus*) due to the same type of fertilization. The conclusion of the study was that three crops absorbed relatively higher amounts of gentamicin (small molecule) than streptomycin (large molecule), and that the levels of antibiotics in plant tissues increased when increasing the antibiotic concentration in manure [64]. The intake of antibiotics into vegetables seedlings was also studied by Ahmed et al. (2015), who showed that cucumber (*Cucumis sativus*), cherry tomato (*Solanum lycopersicum*), and lettuce had relatively high levels of tetracyclines and sulfonamides in the non-edible parts, but lower concentrations in fruit parts and were within acceptable daily intake levels [61].

### 3.2. Products of Animal Origin

The use of antibiotics that may lead to the accumulation of residues in meat, milk, eggs, and honey should not be allowed in foods intended for human consumption. If the use of antibiotics is necessary in the treatment or prevention of various animal diseases, a withholding period must be respected until antibiotic residues are no longer detected [65]. The presence of antibiotic residues in meat from various species of economic interest is considered a significant danger to public health.

The results of the study conducted by Al-Mashhadany in 2020 on sheep meat harvested from supermarkets in Iraq showed that samples contained antibiotic residues at a level higher than the maximum allowed limits. Cooling and freezing, as preservation methods, slightly reduce antibiotic residues in meat. The same study showed that thermal processing of lamb meat (cooking for about 45 min) leads to a transformation of antibiotic residues into inactive residues against bacteria [66].

Another study conducted by Babapour. et al. in 2012 on meat samples collected from Iran obtained similar results in terms of the incidence rate of antibiotic residues in meat [67]. A higher incidence of antibiotic residue has been reported in Nigeria in beef samples [68]. In contrast, the lowest incidence rates were reported in sheep samples analyzed in Spain [48]. The presence of fluoroquinolone residues (enrofloxacin and ciprofloxacin) in some Indonesian chicken samples indicated that it were used by farmers in poultry feed [69,70].

Milk with antibiotic residues significantly influences the technological process of obtaining dairy products, which includes the technology used for dairy yeasts. Antibiotics can get into milk from treatments applied to sick animals or (less often and not recommended) through the use of preservatives. Milk with antibiotic residues is considered a rigged food on the market. By consuming unpasteurized milk, antibiotic-resistant bacteria can be transmitted to consumers, especially in areas with a dense population and a lower degree of development, where there is a risk of improper storage of milk and dairy products [71,72]. Milk and dairy products are exposed to contamination by antibiotics and other drug residues, but also to neutralizing and preserving substances [73,74,75]. Analysis of antibiotic residues in dairy products (pasteurized drinking milk, yogurt, sour cream, whipped milk, cheese) leads to the identification of the gentamicin/neomycin group, especially in sour cream. Macrolides sometimes appear in cheese, and tetracyclines in sour cream and cheese. Here, we discuss about milk samples subject to confirmatory investigations, used for human consumption. Moghadam et al. (2016) identified that 38.5% of raw milk samples collected from the Iranian province of Khorasan Razavi had penicillin residues, while Ghanavi et al. (2013) reported identification of residues of 11% antibiotics of cow milk samples collected from different regions of Iran [76,77]. Studies conducted by Vinu, (2021) showed that there is a direct correlation between the stage of lactation and the presence of antibiotic residues in milk; 34.3% of positive samples came in the lactation stage of 0–70 days, 20% between 70–140 days of lactation, and 45.7% between 140–305 days of lactation [78]. Additionally, Knappstein et al. (2004) highlighted a direct correlation between milk production and the presence of antibiotic residues (cefquinomas), their level not being influenced by the frequency with which milking was performed [79].

In most countries, eggs are the main product generated from backyard poultry production systems due to the fact that they can be quickly consumed or sold to meet essential family needs. In a study by Cornejo et al. (2020), in Chile, the presence of antimicrobial residues in eggs, such as tetracyclines, beta-lactams, aminoglycosides, and macrolides, was analyzed. The survey showed that all samples were positive for at least one of the four antimicrobials tested [80]. Another recent study from China concluded that careful monitoring should be imposed on antibiotic residues in poultry eggs, after detecting 30% positive egg samples for quinolones, tetracyclines, and sulphonamides [81].

The most common contaminations of honey can be explained by treatments in order to control honeybee diseases and contaminants coming from procedures applied in agriculture [82]. The European Union has forbidden the use of antibiotics for bees, this aspect is strictly enforced by recent legislation [83,84]. The most common and important antibiotics found in honey are beta-lactams (penicillin, ampicillin, cloxacillin, amoxicillin for bacterial infections), amphenicols (thiamphenicol, florfenicol, chloramphenicol which are carcinogenic antimicrobials), tetracyclines (oxytetracycline, chlortetracycline, tetracycline for bacterial diseases), macrolides (erythromycin, tylosin, oleandomycin and spiramycin), and aminoglycoside, fluoroquinolones (ciprofloxacin, enrofloxacin, norfloxacin—growth enhancing) [85,86,87].

### 3.3. Methods of Analysis

Analytical techniques for determining antibiotics have gradually evolved with the advent of increasingly advanced technology. If, 50 years ago, the usual technique was based on inhibiting the development of known bacterial cultures, the so-called microtest (the principle used in the antibiogram), today we are see the determination of antibiotics using high performance liquid chromatography coupled with mass spectrometry (LC-MS/MS).

Analysis methods can be classified into screening analysis methods and confirmatory analysis methods.
(a)Screening analysis methods list

Screening methods use equipment that is more readily available in terms of price and mode of operation to identify a group of antibiotics or an antibiotic, with or without quantification of that antibiotic. The general rule is that any result obtained by a screening method must be confirmed using a confirmation method. The screening methods used are the most varied, and are based on different principles, such as microbial inhibition, enzyme immunoassay, stick format (lateral flow devices), radioimmunoassay (RIA), chemiluminescence immunoassay (CLIA), fluorescence immunoassay (FIA), and colloidal gold immunoassay (CGIA). Screening methods for antibiotics have proven to be useful and fast tools that provide results with a high accuracy and sensitivity, which can guarantee safe food [17].
Microbial Inhibition Test (Microtest)

This test is based on the incubation of environmental plates with a suspension of a known concentration of bacterial strains, which is added to the test sample. If the test sample has an antibiotic, it will not allow the development of specific colonies, thus opening a halo area around the sample to be analyzed [88]. This test is an expensive test that involves specific endowments that are specific to a food microbiology laboratory, as well as specialized personnel; another major disadvantage is the obtaining of results after an average of 18 h of incubation, and not in 1–2 h as in other screening methods.
2.Delvotest

This is a classic test for determining antibiotics in milk and is based on the whole principle of microbial inhibition. In the absence of an antibiotic, the bacterial suspension develops and the opacity of the environment, or the change in color due to the appearance of acid in the bacteria-growing activity, is noted. In the presence of an antibiotic, the bacterial strain does not develop and there is an area of inhibition or a lack of environmental color change [89]. This type of test is very sensitive to β-lactam antibiotics, but can also be used for sulfonamides and other antimicrobials. This type of test requires incubation for several hours before results can be visualized, so this test was modified by borrowing principles from the enzyme immunoassay, thus forming antibody–antibiotic complexes that develop a color reaction in the presence an enzyme. A low intensity usually means positive, while a high intensity is considered negative. These tests are more expensive than conventional tests with microbial inhibitors, but provide a result in minutes. The major disadvantage is that they only detect substances that react immunologically with the receptor.
3.Enzyme-linked immunosorbent assay (ELISA)

This method is based on the classical antigen–antibody reaction in the presence of a conjugate. The technique of obtaining the antibody is relatively simple and is based on the body’s ability to generate antibodies to a particular antigen, usually using different adjuvants that increase the body’s ability to produce antibodies [90]. This assay represents the most common screening test for detecting of antibiotic residues, especially in food samples [91], and the sensitivity of this method is sometimes superior to confirmatory methods [92]. Depending on the antibiotic detected, different immunoenzymatic techniques have emerged. Thus, for the determination of fluoroquinolones, competitive ELISA methods have been developed using the reaction between the antigen–conjugate (fluoroquinolone)–antibody (bovine serum albumin) and the specific polyclonal antibody [93]. Additionally, for the determination of chloramphenicol and tetracycline, the specific monoclonal antibody is used in most cases [94,95]. ELISA can also assess multiple residues of antibiotics in different foods, so a new colorimetric and dual-colorimetric ELISA test has been developed for simultaneously determination of 13 fluoroquiunolone residues and 22 sulfonamides [96,97]. The sensitivity of the ELISA method has been improved by the addition of a biotin-streptavidin compound that allows better catalysis of the substrate [98,99,100]. Immunoenzymatic analysis techniques have been developed for the determination of antibiotics in both animal products (milk, eggs, meat) and aquaculture products and feed [101,102]. In conclusion, it can be said that the enzyme-linked immunosorbent assay is a fast, sensitive, and easy to implement test [100], but that it also has some disadvantages, including a fairly high percentage of false positive results due to cross-reactions and low reproducibility [95].
4.Radioimmunotest (RIA)

This technique uses isotope-labeled, as well as unlabeled, antigens to react competitively with antibodies. This technique is used to detect antibiotics in various products of animal origin, as well as in various products in the aquatic environment [103].
5.Chemiluminescence Immunoassay (CLIA)

This test is widely used due to the fact that it is an easy, fast, sensitive, and selective test [104]. CLIA is based on the combination of two systems, namely the immune response and actual chemiluminescence analysis. Due to its high specificity and sensitivity, CLIA is used in many fields, but it has limitations due to the compounds used such as acridinium derivatives and the immediate emission of light is a disadvantage due to its measurement problems [105].
6.Colloidal gold immunochromatographic assay (CGIA)

This new test uses colloidal gold as a tracer in an alkaline environment, which interacts with negatively and positively charged groups and antibody protein molecules [106]. This technique has been developed for the rapid determination of chloramphenicol [107]. Another CGIA technique was developed for the simultaneous determination of quinolones, tetracycline, and sulfonamide in milk, thus allowing the concomitant determination of 36 different antibiotics in less than 10 min [108]. A high-sensitivity CGIA test was developed for the determination of streptomycin in pig milk and urine, with a very low limit of detection of 2.0 ng/mL for milk and 1.9 ng/mL in urine [99].
7.Fluorescence Polarization Assay (FPIA)

The principle of the test is competitive and is based on the binding of fluoroflora to a specific antigen and highlights the fluorescent compound as a standard compound needed to detect and identify an unknown antigen. If the antibiotic sought is not in the sample, a tracer will be bound to the antibody and the signal will be high [109]. Various FIA techniques have been developed for the simultaneous determination of several fluoroquinolones in food; these techniques are based on the use of monoclonal antibodies. Other FPIA techniques have been developed for the simultaneous determination of cephalexin and cefadroxil in milk samples, gentamicin in goat’s milk, as well as other antibiotic [99,110]. FIA is an easy-to-implement screening method that allows the simultaneous detection of various antibiotics in a short period of time. As a disadvantage, this test requires a sample preparation step to extract the antibiotic from the sample, as well as a filtering step to obtain a colorless sample that does not affect the reading of the sample relative to the fluorescence points [111].
8.Lateral flow immunoassay (LFIA)

Until a few years ago, this type of test had applications only in areas such as the diagnosis of various diseases, pregnancy, and identification of various toxins in the environment. However, recently, there have been applications for the use of LFIA for the simultaneous detection of beta-lactams, quinolones, sulfonamides, and tetracyclines in food [112]. The advantages of using this test include its ease of use, increased shelf life—up to 2 years, and use at room temperature. Like any very simple test, it has many disadvantages: many false positive or false negative results, low reproducibility, etc. [113].
(b)Confirmatory analysis methods

Analysis techniques have evolved gradually and the need for more and more advanced methods has been a natural consequence. While screening methods, with the exception of microtest, do not involve major costs or specialized personnel, confirmation methods involve the use of expensive equipment (LCMSMS, GCMSMS) and highly qualified personnel. Depending on the antibiotics of interest, the equipment and extraction steps are different. A mass spectrometer (MS) is an equipment with an operating principle that is the production of ions, their sorting according to the specific mass-to-load ratio (*m*/*z*), and the analysis of the obtained signals. Each compound is characterized by a specific *m*/*z* ratio, with data present in the literature, as well as in the software of the latest generation of equipment, which comes equipped with data libraries that allow the identification of compounds against reference values [114]. The use of MS is very common in the analysis of antibiotic residues because it has a much higher specificity than screening methods and allows the simultaneous determination of many classes of antibiotics. GCMS/MS has previously been used, but due to the fact that the processing of samples for gas chromatography is more cumbersome and often requires derivatization steps for signal amplification, the development of methods of analysis for the simultaneous determination of antibiotics of several classes is more difficult [115].
Liquid chromatography coupled with mass spectrometry (LC/MS/MS)

Liquid chromatography coupled with mass spectrometry is a commonly used technique for determining antibiotic residues in food. This technique is used by both official laboratories for routine analyses as well as by national reference laboratories and European reference laboratories. With liquid chromatography coupled with mass spectrometry, determining seven classes of antibiotics was possible, totaling 30 antibiotics in less than 8 min. The preparation of a sample involves the weighing of 1 g of sample of a meat obtained and is evaporated and then taken up again with 1 mL of ultrapure and used for introduction into the LC/MS/MS. LC/MS/MS equipment which is optimized for the identification and quantification of each compound of interest [114]. Another method of determining antibiotics allows the determination of 46 antibiotics from different classes. This method was developed for the determination of cows from cow’s milk, beef, sheep, pigs, equines, and birds, as well as fish and shrimp [116]. As a general principle, a sample is extracted with a mixture of solvents, purified by passing through an SPE column, and then injected into the LC/MS/MS. For antibiotic residues, the method of analysis must meet the performance criteria of European Commission Decision No. 2002/657 [117]. In general, regardless of the laboratory in which antibiotic residues are determined by LC/MS/MS, the extraction protocol is generally the same; namely, extraction of the sample with an organic solvent, purification by passing through an SPE column, injection into LC/MS/MS, and the criteria for performance must comply with European Commission.
2.Gas chromatography coupled with mass spectrometry (GC/MS/MS)

Using gas chromatography coupled with mass spectrometry, there are fewer applications because the derivatization stage is cumbersome and affects the long-term life of the equipment, so applications are restricted to 1–2 classes of antibiotics that can be determined simultaneously. The sample preparation protocol is generally the same as for LCMS/MS; namely, sample extraction with organic solvent, purification by passing through an SPE column, specific sample derivatization, injection into GC/MS/MS, and performance criteria must meet European Commission Decision No. 2002/657.

Official laboratories in Romania use methods that use LC/MS/MS equipment and analyzed compounds are those provided by the Surveillance and Control Program in the field of food safety; the developed methods are based on standardized methods or are provided by European reference laboratories. Thus, in Romania a method for determining 14 classes of antibiotics totaling 83 antibiotics is used, and tissue samples for milk, eggs, and honey are analyzed. Methods for LCMS/MS determination of chloramphenicol, nitrofurans, nitroimidazole in food stuffs of animal origin are also developed.

The methods of analysis used to identify antibiotic residues have advantages and disadvantages, as shown in Table 3.

## 4. Impact of Food Consumption with Antibiotic Residues on Consumers’ Health

The concept of “One Health”, promoted by the World Health Organization and the World Organization for Animal Health, emphasize the idea of an intimate relationship between humans, animals, and the environment, all leading to a unique concept of health. Thus, the entire chain must be considered in order to maintain equilibrium, carefully using medicinal products at all three levels [20]. Today, there is major public awareness about the consequences of prolonged and increased use of antibiotics in animal livestock production [118]. Microorganisms have the ability to develop antibiotic-resistant genes, resulting in increased survival, thus minimizing treatment options for microbial infections and leading to increased mortality among humans [119,120,121,122]. Foods from animal origins are considered key reservoirs of antibiotic residues, which occur as a result of the use of antibiotics industrially, thus contributing to the induction of globally antibiotic resistance. Antibiotic-resistant bacteria have been identified in animal-origin food products, in feeds, as well as in humans. Globally, very large differences have been reported between geographical areas in terms of prevalence of animal-origin antibiotic-resistant bacteria and antibiotic-resistance genes [123]. According to recent studies, antibiotics remain in animal-origin food products, such as milk, meat, and eggs, even after heat treatment, and lead to the development of gastrointestinal disorders and allergies in humans, or even the appearance of antibiotics-resistant superbugs. Furthermore, along with antibiotic resistance, the ineffectiveness of antibiotic therapy for human treatments is increasing [22]. There are a number of studies that mention that the continuous and abusive use of antibiotics frequently causes the development of antibiotic-resistant bacteria. Moreover, multidrug-resistant bacterial infections can progressively increase mortality, thus posing a threat to public health [124]. The major problem frequently mentioned by researchers is that, over time, bacteria may adapt and acquire resistance to active phenolic components similar to antibiotics; therefore, their use must also be taken into account [8]. Kumar et al. (2020) reviewed this topic in order to better understand the mechanisms of development and dissemination of antibiotic resistance genes in nutritional, clinical, agricultural and environmental contexts [125,126,127]. In the same study, other dietary strategies were considered to replace medicated feed with probiotics, essential oils, or antibodies, with a preventive role against bacterial infections. A solution to antibiotic resistance needs efforts from several fields of activity, including agriculture, but also veterinary medicine (microbiology, biochemistry, medical clinic, and genetics), by replacing medicated feed with alternative therapies [124,128,129]. Animal husbandry is a key component of the global economy, and is a major contributor to food provision. In order for animals to gain weight, they receive medicated feed containing antibiotics or antibiotics are introduced into drinking water. Antibiotics are introduced into farm animal feed, even for preventive purposes. Allen mentioned (2014) that this activity leads to massive accumulation of antibiotics in the environment, and subsequently leads to the acquisition of antibiotic resistance by microorganisms [130]. Several studies have indicated that the spread of antibiotic-resistant microorganisms in humans is mainly due to the consumption of animal-origin foods and beverages contaminated with antibiotic residues, or through the consumption of water contaminated by environmental pollution [25,124,131].

Therefore, in order to reduce antibiotic-resistant bacterial infections worldwide, measures regarding the use of antibiotics for non-therapeutic purposes, such as the use of antibiotics in animals feed, have been taken, when the products are intended for human consumption. Banning the use of avoparcin in animal feed in the European Union has reduced the incidence of antibiotic resistance in animals, and thus its occurrence in humans [124,132]. Antibiotic resistance is a global problem that affects public health, with socio-economic repercussions and it significantly influences the use of antibiotics in animal feed of economic interest. The development of the WHO Global Action Plan and the FAO Global Action Plan, in line with the One Health concept, is a requirement to prevent the transmission of antibiotic resistance, from farm to fork [123].

Every year, 33,000 people die as a direct consequence of infections caused by antibiotic-resistant bacteria, a number comparable to the passengers of more than 100 medium-sized aircraft [133].

Many classes of antibiotics have been recognized, and their use should be limited. In December 2019, the EMA classified antimicrobials into four categories, A to D (Figure 1).

The irrational use of antibiotics in animal husbandry and subsequent pollution of the environment (wastewater, animal- and non-animal-origin food products, soil in places where manure was applied), inevitably leads to the formation of antibiotic resistance. Erofeeva et al. (2021) mentioned that the problem is that scientists have not discovered any new group of antibiotics since 2004 [25] (Figure 2).

In the European Union, the provisions of European Regulation 37/2010 concerning maximum permitted limits by product categories and by food producing species are applicable (Table 4).

Improper use of antibiotics in food-producing animals contributes to the appearance of antibiotic resistance. WHO recommends that farmers and the food industry stop using antibiotics routinely to promote growth and prevent disease in healthy animals. Recommendations aim to help preserve the effectiveness of antibiotics that are important for human medicine by reducing their unnecessary use in animals. WHO declares that, in different countries, approximately 80% of total antibiotic consumption appears in the animal sector, largely to promote the growth of healthy animals. Overuse and misuse of antibiotics in animals and humans is contributing to the rising threat of antibiotic resistance. Some types of bacteria that cause serious infections in humans have already developed resistance. WHO strongly recommends a reduction in use of all classes of antibiotics for food-producing animals, including a complete restriction of these antibiotics for growth promotion and disease prevention without a diagnosis. “Scientific evidence demonstrates that overuse of antibiotics in animals can contribute to the emergence of antibiotic resistance” says Dr. Kazuaki Miyagishima, Director of the Department of Food Safety and Zoonoses at WHO. Many countries have already taken actions to reduce the use of antibiotics in food-producing animals. For example, since 2006, the European Union has banned the use of antibiotics for growth promotion. In addition, consumers promote the marketing of meat raised without the routine use of antibiotics, with some major food chains adopting “antibiotic-free” policies for meat supplies. The general objective is to encourage the prudent use of antibiotics in order to slow down antimicrobial resistance and maintain the effectiveness of antibiotics for medicine [136]. About two-thirds of U.S. antibiotics that are important to people are sold for use in food animal production. Yet, experts have long warned of the public health threat from regularly exposing herds of thousands, even tens of thousands, of animals to human-class antibiotics in their feed for prolonged periods of time. The U.S. Food and Drug Administration (FDA) lists 89 medically important antibiotics currently added to animal feeds. According to the www.nrdc.org website (accessed on 6 May 2022), since 2016, the FDA has flagged the lack of clear antibiotics use limits as a significant problem needing a remedy, thus, at this moment in the USA, there are no clear limits for the use of antibiotics. The same site mentions the deadline for establishing and publishing these data as 2023 [137].

Numerous studies from different parts of the world mention that antibiotic residues in feed stuffs are, at present, a large problem, and can lead to major associated health problems, including antibiotic resistance, toxicity, hypersensitivity reactions, teratogenicity, and carcinogenicity (Figure 3) [1].

Most antibiotics cause side effects in humans (Table 5) [134].

A large number of studies refer to the management of antibiotic residues, antibiotic-resistant bacteria (ARB), and antibiotic resistance genes (ARG) can be found in dairy manure and may contribute to the spread of antibiotic resistance (AR). More than 60 ARGs can be found in milk manure (including β-lactam and tetracycline resistance genes), although correlations with antibiotic use, residues, and ARBs have been inconsistent, possibly due to sampling and analytical limitations. Antibiotic resistance genes often persist through these systems, although optimal management and a higher operating temperature may facilitate their attenuation [138,139,140]. Elements related to the mechanism of action and resistance in humans are presented in Table 6 [141] (after Iwu et al., 2020).

Worldwide, and under the auspices of the World Health Organization (WHO), lectures and interactive broadcasts are organized to emphasize the importance of judicious administration of antibiotics in both human and veterinary use, or in agriculture. People should be aware of the need for antibiotics to be used correctly and to reduce the abuse of antibiotics. In addition, national programs aimed at the screening of antibiotic residues in various types of food are being continuously updated.

## 5. Conclusions

The issue of the presence of antibiotic residues in food is intensely debated. Numerous research studies have highlighted the irrational use of antibiotics and the risk of problems with consumer antibiotic resistance, spread by foods with antibiotic residues.

The concentration and type of antibiotic found in the form of residues varies depending on the geographical area and the type of food analyzed. Available studies present antibiotic residues in all food groups: meat and meat products, milk and dairy products, eggs, honey, and non-animal-origin products.

Although alarm signals are drawn regarding irrational antibiotic use, exceeding applicable legal requirements are identified. While the European Union has clearly established limits for antibiotic residues, in the United States, legislation does not include such values, with a deadline set for 2023 to draw up these legislative requirements. Additionally, in recent years, awareness of the irrational use of antibiotics programs have been launched, but still it is necessary to develop these in order to increase the understanding of producers and consumers regarding the use of antibiotics. These measures should be implemented worldwide.

## Figures and Tables

**Figure 1 foods-11-01430-f001:**
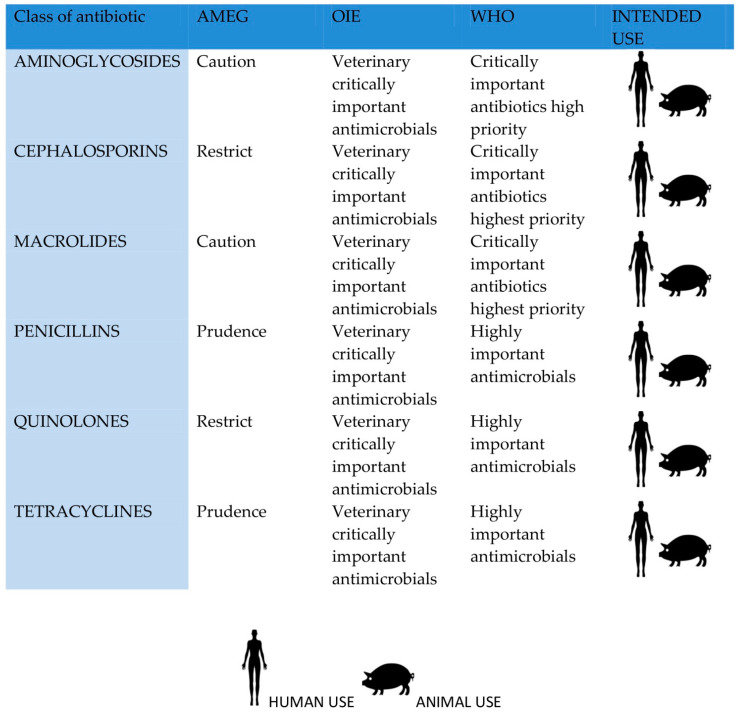
Comparison listing of CIAs by WHO (2019), OIE (2018), and AMEG—EMA (2019) for the major classes of antibiotics adapted from [134].

**Figure 2 foods-11-01430-f002:**
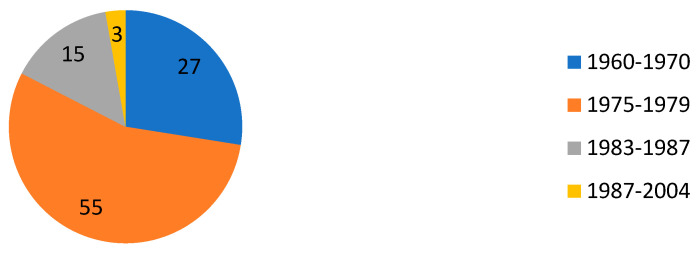
Discovery of antibiotics adapted from [25].

**Figure 3 foods-11-01430-f003:**
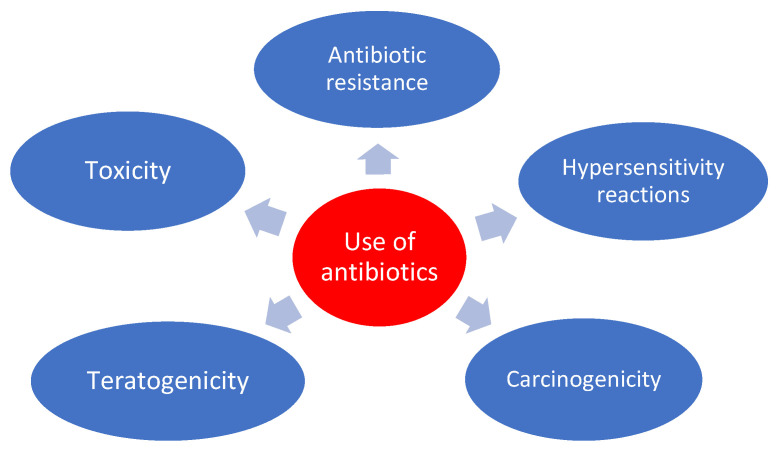
Effects of antibiotics use on human health.

**Table 1 foods-11-01430-t001:** Bibliometric analysis according with “Web of Science” database.

Field of Research	Number of Scientific Publications
Antibiotic residues in food	67
Antibiotic residues in feed	17
Antibiotic residues in animal origin food	12
Antibiotic residues in non-animal origin food	6
Antibiotic residues in meat	54
Antibiotic residues in meat products	4
Antibiotic residues in fish	12
Antibiotic residues in milk	292
Antibiotic residues in eggs	21
Antibiotic residues in honey	45

**Table 2 foods-11-01430-t002:** Presence of varying concentrations of antibiotic residues in the different animal-derived products in some developing countries. Reprinted from [37].

Antibiotic Residue	Concentration	Sample	Country	Literature
Oxytetracycline		Chicken	Tanzania	Kimeria et al. [38]
	2604.1 ± 703.7 µg/kg	Muscle		
	3434.4 ± 604.4 µg/kg	Liver		
	3533.1 ± 803.6 µg/kg	Kidney		
		Beef	Nigeria	Olufemi and Agboola [39]
	51.8 ± 90.53 µg/kg	Muscle		
	372.7 ± 366.8 µg/kg	Kidney		
	1197.7 ± 718.9 µg/kg	Liver		
		Cattle	Ethiopia	Bedada et al. [40]
	15.92 to 108.34 µg/kg	Muscle		
	9.02 to 112.53 µg/kg	Kidney		
Enrofloxacin	0.73 and 2.57 µg/kg	Chicken tissues	Iran	Tavakoli et al. [41]
Chloramphenicol	1.34 and 13.9 µg/kg			
Penicillin	0.87 and 1.3 µg/kg	Calves muscles		
Oxytetracycline	3.5 and 4.61 µg/kg			
Quinolones	30.81 ± 0.45 µg/kg	Chicken	Turkey	Er et al. [42]
	6.64 ± 1.11 µg/kg	Beef		
Tetracyclines		Chicken	Egypt	Salama et al. [43]
	124 to 5812 µg/kg	Breast		
	107–6010 µg/kg	Thigh		
	103 to 8148 µg/kg	Livers		
		Chicken	Cameroon	Guetiya-Wadoum et al. [44]
	150 ± 30 µg/g	Liver		
	62.4 ± 15.3 µg/g	Muscle		
		Beef	Kenya	Muriuki et al. [45]
	50 to 845µg/kg	Kidney		
	50 to 573 µg/kg	Liver		
	23–560 µg/kg	Muscle		
Amoxicillin	9.8 to 56.16 µg/mL	Milk	Bangladesh	Chowdhury et al. [46]
	10.46 to 48.8 µg/g	Eggs		
Sulfonamides	16.28 µg/kg	Raw milk	China	Zheng et al. [47]
Quinolones	23.25 µg/kg			
Oxytetracycline	199.6 ± 46 ng/g	Beef	Zambia	Nchima et al. [48]
Sulphamethazine	86.5 ± 8.7 ng/g			
Penicillin G	15.22 ± 0.61 µg/L	Fresh milk	Nigeria	Olatoye et al. [49]
	7.60 ± 0.60 µg/L	Cheese (wara)		
	8.24 ± 0.50 µg/L	Fermented milk (nono)		
Sulphonamides		Chicken	Malaysia	Cheong et al. [50]
	0.08–0.193 µg/g	Liver		
	0.006–0.062 µg/g	Breast		
Tetracycline	>0.1 µg/mL	Raw milk	India	Kumari Anjana et al. [51]
Oxytetracycline				
Sulfadimidine				
Sulfamethoxazole				

**Table 3 foods-11-01430-t003:** Advantages and disadvantages of different analysis methods.

Analytical Method	Advantages	Disadvantages
Screening analysis methods	easy to operate	mainly qualitative methods
	low price	any result obtained by a screening method must be confirmed by a confirmation method
(a) Microbial inhibition test (microtest)	specificity–if the test sample has an antibiotic, it will not allow the development of specific colonies, thus opening a halo area around the sample to be analyzed	expensive test that involves specific endowments specific to a food microbiology laboratory as well as specialized personnel
		obtaining of results only after an average of 18 h of incubation
(b) Delvotest	classic test for determining antibiotics in milk	more expensive than conventional tests
	very sensitive to β-lactam antibiotics	detects only substances that react immunologically with the receptor
(c) Enzyme-linked immunosorbent assay (ELISA)	sensitivity of this method is sometimes superior to confirmatory methods	fairly high percentage of false positive results due to cross-reactions
	used for the multi-residue determination of antibiotics in different foods	low reproducibility
	fast, sensitive and easy to implement test	
(d) Radioimmunotest (RIA)	high selectivity	high concentrations of other molecules with antibody affinity could inactivate it
	high sensitivity	
(e) Chemiluminescence immunoassay (CLIA)	easy, fast, sensitive and selective test	measurement problems due to the compounds used, such as acridinium derivatives and the immediate emission of light
(f) Colloidal gold immunochromatographic assay (CGIA)	rapid determination of chloramphenicol	high price
	simultaneous determination of quinolones, tetracycline and sulfonamide in milk; 36 different antibiotics in less than 10 min	
(g) Fluorescence polarization assay (FPIA)	easy-to-implement screening method that allows the simultaneous detection of various antibiotics in a short period of time	requires a sample preparation step to extract the antibiotic from the sample
		a filtering step to obtain a colorless sample that does not affect the reading of the sample relative to the fluorescence points
(h) Lateral flow immunoassay (LFIA)	ease of use	many false positive or false negative results,
	increased shelf life—up to 2 years at room temperature	low reproducibility
Confirmatory analysis methods	higher specificity than screening methods	use of expensive equipment
	allows the simultaneous determination of many classes of antibiotics	super qualified personnel
(a) Liquid chromatography coupled with mass spectrometry (LC/MS/MS)	Determination of 7 classes of antibiotics, 30 antibiotics in less than 8 min	high price
	method of analysis must meet the performance criteria of European Commission Decision No. 2002/657	
(b) Gas chromatography coupled with mass spectrometry (GC/MS/MS)	standardized methods or provided by European reference laboratories	applications are much lower because the derivatization stage is cumbersome and affects the long-term life of the equipment, so the applications are restricted to 1–2 classes of antibiotics that can be determined simultaneously

**Table 4 foods-11-01430-t004:** Maximum residues limits of antibiotic in products of animal origin marketed in the European Community [135].

Active Substance	Animal Species	Target Tissue	MRL
Amoxicillin	All food-producing species	Muscle Fat Liver Kidney Milk	50 μg/kg 50 μg/kg 50 μg/kg 50 μg/kg 4 μg/kg
Ampicilin	All food-producing species	Muscle Fat Liver Kidney Milk	50 μg/kg 50 μg/kg 50 μg/kg 50 μg/kg 4 μg/kg
Avilamycin	Porcine, poultry, rabbit	Muscle Fat Liver Kidney	50 μg/kg 100 μg/kg 300 μg/kg 200 μg/kg
Bacitracin	Bovine	Milk	100μg/kg
Benzylpenicillin	All food-producing species	Muscle Fat Liver Kidney Milk	50 μg/kg 50 μg/kg 50 μg/kg 50 μg/kg 4 μg/kg
Cefacetrile	Bovine	Milk	125 μg/kg
Cefapirin	Bovine	Muscle Fat Kidney Milk	50 μg/kg 50 μg/kg 100 μg/kg 60 μg/kg
Cefazolin	Bovine, ovine, caprine	Milk	50 μg/kg
Chlortetracycline	All food- producing species	Muscle Liver Kidney Milk Eggs	100 μg/kg 300 μg/kg 600 μg/kg 100 μg/kg 200 μg/kg
Clavulanic acid	Bovine, porcine	Muscle Fat Liver Kidney	100 μg/kg 100 μg/kg 200 μg/kg 400 μg/kg
Cloxacillin	All food-producing species	Muscle Fat Liver Kidney Milk	300 μg/kg 300 μg/kg 300 μg/kg 300 μg/kg 30 μg/kg
Colistin	All food-producing species	Muscle Fat Liver Kidney Milk Eggs	150 μg/kg 150 μg/kg 150 μg/kg 200 μg/kg 50 μg/kg 300 μg/kg
Cloxacillin	All food-producing species	Muscle Fat Liver Kidney Milk	300 μg/kg 300 μg/kg 300 μg/kg 300 μg/kg 30 μg/kg
Dicloxacillin	All food-producing species	Muscle Fat Liver Kidney Milk	300 μg/kg 300 μg/kg 300 μg/kg 300 μg/kg 30 μg/kg
Doxycycline	Bovine Porcine, poultry	Muscle Liver Kidney Not for use in animals from which milk is produced for human consumption	100 μg/kg 300 μg/kg 600 μg/kg
		Muscle Skin and fat Liver Kidney Not for use in animals from which eggs is produced for human consumption	100 μg/kg 300 μg/kg 300 μg/kg 600 μg/kg
Enrofloxacin	Bovine, ovine	Muscle Fat Liver Kidney Milk	100 μg/kg 100 μg/kg 300 μg/kg 200 μg/kg 100 μg/kg
Enrofloxacin	Porcine, rabbit	Muscle Fat Liver Kidney	100 μg/kg 100 μg/kg 200 μg/kg 300 μg/kg
	Poultry	Muscle Skin and fat Liver Kidney	100 μg/kg 100 μg/kg 200 μg/kg 300 μg/kg
	All other food-producing species	Muscle Skin and fat Liver Kidney	100 μg/kg 100 μg/kg 200 μg/kg 200 μg/kg
Erythromycin A	All other food-producing species	Muscle Fat Liver Kidney Milk Eggs	200 μg/kg 200 μg/kg 200 μg/kg 200 μg/kg 40 μg/kg 150 μg/kg
Gentamicin	Bovine, porcine	Muscle Fat Liver Kidney Milk	50 μg/kg 50 μg/kg 200 μg/kg 750 μg/kg 100 μg/kg
Kanamycin A	All food-producing species except fin fish	Muscle Fat Liver Kidney Milk	100 μg/kg 100 μg/kg 600 μg/kg 2500 μg/kg 150 μg/kg
Lincomycin	All food-producing species	Muscle Fat Liver Kidney Milk Eggs	100 μg/kg 50 μg/kg 500 μg/kg 1500 μg/kg 150 μg/kg 50 μg/kg
Marbofloxacin	Bovine, porcine	Muscle Fat Liver Kidney	150 μg/kg 150 μg/kg 50 μg/kg 150 μg/kg
Neomycin B	All food-producing species	Muscle Fat Liver Kidney Milk Eggs	500 μg/kg 500 μg/kg 500 μg/kg 5000 μg/kg 1500 μg/kg 500 μg/kg
Oxacillin	All food-producing species	Muscle Fat Liver Kidney Milk	300 μg/kg 300 μg/kg 300 μg/kg 300 μg/kg 30 μg/kg
Oxytetracycline	All food-producing species	Muscle Liver Kidney Milk Eggs	100 μg/kg 300 μg/kg 600 μg/kg 100 μg/kg 200 μg/kg
Streptomycin	All ruminants, porcine, rabbit	Muscle Fat Liver Kidney	500 μg/kg 500 μg/kg 500 μg/kg 1000μg/kg
Sulfonamides	All food-producing species	Muscle Fat Liver Kidney	100 μg/kg 100 μg/kg 100 μg/kg 100 μg/kg
Tylosin A	All food-producing species	Muscle Fat Liver Kidney Milk Egg	100 μg/kg 100 μg/kg 100 μg/kg 100 μg/kg 50 μg/kg 200 μg/kg

**Table 5 foods-11-01430-t005:** Possible effects due to antibiotics in human.

Group of Antimicrobials	Main Effects	Clinical Signs
Sulphonamides	Skin reactions	Mild rash to severe toxidermia are some of the skin reactions following human exposure to sulphonamide
	Hypersensitivity mentioned averse reactions reactions	Contact sensitization confirmed for topical medicinal products
	Blood dyscrasias	Hemolytic anemia, neutropenia, thrombocytopenia and pancytopenia
	Carcinogenicity (thyroid)	Sulfamethazine dose-dependent increase in follicular cells adenomas of thyroid gland
Penicillins	Hypersensitivity reactions	Association with IgE-mediated allergic anaphylaxis 10% of the human population is believed to be allergic
	Anaphylaxis	Human reaction based on penicilloyated (amoxicilloyated) residues in milk and meat. Amoxicillin (AX), with or without clavulanic acid, is the most common elicitor of allergy. Very low levels (6 μg/L) can cause this reaction; therefore, especially for milk low MRLs (4 μg/kg) were established for the group of penicillins by EMA and JECFA (Codex). USA—zero tolerance for residues in milk
	Influence of starter cultures in food processing	Sufficient evidence that consumption of beef or pork containing residues of penicillins exceeding MRLs causing anaphylactic reactions
Tetracyclines	Possible influence of human intestine microbiome	MRLs set based on the microbiological ADI. In the period of EMA assessment, it was concluded that there is no induction of resistant enterobacteria at the dose 2 mg per person per day—on the other hand, in an in vitro study to assess the impact of tetracycline on the human intestinal microbiome, there was screened the variability of the presence of tet genes after exposure of low concentrations 0.15, 1.5, 15 and 150 μg/mL of tetracycline, after 24 h and 40 days and variable to slight increase of the tetracycline gene copies occurred.

**Table 6 foods-11-01430-t006:** Mechanism of action and resistance mechanism of antibiotics in human. Adapted from [141]. (after Iwu et al., 2020).

Antibiotics Class	Example (s)	The Mechanism(s) of Action	Resistance Mechanism(s)
β-lactams	Cephalosporins, Penicillins, Cefotaxime, Monobactams, Carbapenems	Cell wall biosynthesis inhibition	Cleavage by β-lactamases, ESBLs, Carbapenemases, Cefotaximases, and altered Penicillin-binding proteins
Aminoglycosides	Gentamicin, streptomycin	Protein synthesis inhibition	Ribosomal mutations, enzymatic modification, 16S rRNA methylation, and efflux pumps
Phenicols	Chloramphenicol	Inhibition of protein synthesis	Mutation of the 50S ribosomal subunit, reduced membrane permeability, and elaboration of chloramphenicol acetyltransferase
Macrolides	Erythromycin, azithromycin	Alteration of protein synthesis	Ribosomal methylation
Tetracyclines	Minocycline, tigecycline	Alteration of translation	Mainly efflux
Rifamycins	Rifampin	Alteration of transcription	Altered β-subunit of RNA polymerase
Glycopeptides	Vancomycin, teicoplanin	Alteration of cell wall biosynthesis	Altered cell walls, efflux
Quinolones	Ciprofloxacin	Alteration of DNA synthesis	Efflux, modification, target mutations
Streptogramins	Synercid, streptogramin B	Alteration of cell wall biosynthesis	Enzymatic cleavage, modification, efflux
Oxazolidinones	Linezolid	Alteration of formation of 70S ribosomal complex	Mutations in 23S rRNA genes followed by gene conversion
Lipopeptides	Daptomycin	Depolarization of cell membrane	Modification of cell wall and cell membrane

## Data Availability

Not applicable.

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
