# Peer review of "Antibiotic Use in Livestock and Residues in Food—A Public Health Threat: A Review"

_foods, 2022, doi:10.3390/foods11101430_

Round 1

Reviewer 1 Report

Antibiotics residues in foods have been one of important topic in food safety. This review focuses in antibiotic use in livestock, antibiotic residues in food and the methods of detection and impact of food consumption with antibiotic residues on consumers’ health.

  1. The cutting-edge of antibiotics residues in foods should be reviewed such as the influence of antibiotics residues in foods on the genetic drift of drug-resistant bacteria.
  2. How to prevent the antibiotics abuse in foods and how to remove the antibiotics residues in foods and keep the food safety.

Reviewer 2 Report

Ghimpeteanu et al. have reviewed the antibiotic residues in food covering antibiotic use in livestock, antibiotic residues in food products of non-animal/animal origin, method of analysis, and health impacts. Although it is a topic of interest to be reviewed, the authors should be more elaborate especially in terms of providing some essential tables and figures.

  1. The authors should consider revising the title of the review, as the review covers a much wider scope than just public health threat.
  2. A bibliometric analysis using “Web of Science” database should be done to get the data on number of papers published on antibiotics residues and highlight the increasing importance over the years.
  3. Why the legislations and limits imposed by leading authorities like FDA, WHO etc. are not covered in this review?
  4. A table showcasing the different analytical methods, their advantages and disadvantages should be included.
  5. For section 4, a figure showing classification of different possible health effects should be included. Also, contributions of each in terms of a pie chart should be added.
  6. Why the authors have categorically avoided including figures? Besides the suggestions given above, some more figures from the recent publications should be included that would project the public health threat in terms of antibiotic use in livestock, antibiotic residues in food products of non-animal/animal origin, method of analysis, and health impacts.
  7. A table showing different classes of antibiotics, their residue levels, recommended limits etc. should be included.

Reviewer 3 Report

1-The author are not the solely researchers that are studying AR. Several recent publications have been omitted in the manuscript. Authors should add some references to demonstrate their awareness on international research done in various parts of the world as they have mentioned in the abstract.

2-Authors should refer in the abstract that they will describe the methods of analysis of AB in foods. 

3-The conclusion should be more comprehensive. Authors should state some recommendations for future research implications. 

Author Response

Response to Reviewer 3 Comments

1-The author are not the solely researchers that are studying AR. Several recent publications have been omitted in the manuscript. Authors should add some references to demonstrate their awareness on international research done in various parts of the world as they have mentioned in the abstract.

Response 1: – insert in Introduction

Xiuru et al in 2021 and Fuyang et al in 2020, mention again that antibiotics have become widespread in the environment due to their extensive and long-term use, influencing both human health and the system, due to the emergence of antibiotic resistance.

Saeed et al. in 2020, acknowledges the contamination of food with antibiotic residues, mentioning that it is a global problem, due to the improper use of antibiotics. As we have described, the authors of this paper also mentioned methods of analysis to identify antibiotics in food.

  1. Xiuru Yang, Zhi Chen, Wan Zhao, Chunxi Liu, Xiaoxiao Qian, Ming Zhang, Guoying Wei, Eakalak Khan, Yun Hau Ng, Yong Sik Ok, 2021. Recent advances in photodegradation of antibiotic residues in water. Chemical Engineering Journal, Volume 405, 1 February 2021, 126806.
  2. Fuyang Huang, Ziyi An, Michael J. Moran, Fei Liu, 2020. Recognition of typical antibiotic residues in environmental media related to groundwater in China (2009−2019). Journal of Hazardous Materials, Volume 399, 122813. https://doi.org/10.1016/j.jhazmat.2020.122813
  3. Saeed Ahmed, Jianan Ning, Dapeng Peng, Ting Chen, Ijaz Ahmad, Aashaq Ali, Zhixin Lei, Muhammad Abu bakr Shabbir, Guyue Cheng & Zonghui Yuan, 2020. Current advances in immunoassays for the detection of antibiotics residues: a review. FOOD AND AGRICULTURAL IMMUNOLOGY, VOL. 31, NO. 1, 268–290, https://doi.org/10.1080/09540105.2019.1707171.

2-Authors should refer in the abstract that they will describe the methods of analysis of AB in foods. 

Response 2: – insert in abstract

We also aim to present the methods of analysis currently used to determine antibiotic residues in food, methods that are characterized by the speed of obtaining results, or by the possibility of identifying very small amounts of residues.

3-The conclusion should be more comprehensive. Authors should state some recommendations for future research implications. 

Response 3: – insert in conclusions

Although alarm signals are drawn about irrational antibiotic use, exceeding of applicable legal requirements are identified. If in the European Union there are clearly established limits for antibiotic residues, in the United States, the legislation does not include such values, being set the deadline of 2023 to drawn up these legislative requirements.

Also in recent years awareness of the irrational use of antibiotics programs have been launched, but still it is necessary to develop them in order to increase the perception of producers and consumers on the use of antibiotics. These measures should be implemented worldwide.

Reviewer 4 Report

The manuscript proposes an interesting picture of the state of the art regarding the antibiotics use food industry and their residual presence in foods. The writing should be revised since it is poor and, in some parts, not clear: see lines 71-74, line 107, etc. A table reporting the values observed in different crops and foods would be useful to assess the title issue and end points of the manuscript. Appropriate References should added to give clear information regarding the sources and values observed in different foodstuff. Regulations existing, accepted limits of use and differences in the different Countries should be better put in evidence. Examples of Methods of analysis applications and limits should be given, and the results reported commented and assessed in the text. The Conclusion section should be be dedicated to the end points, contribution to the field and perspective view of the the Authors with reference to the topic of the proposed review paper.

Round 2

Reviewer 1 Report

It is ok.

Reviewer 2 Report

The authors have satisfactorily addressed all the comments raised by reviewers.